# Terahertz Attenuated Total Reflection Spectral Response and Signal Enhancement via Plasmonic Enhanced Sensor for Eye Drop Detection

**DOI:** 10.3390/s23198290

**Published:** 2023-10-07

**Authors:** Eugene Soh Jia Hao, Nan Zhang, Qiang Zhu, Xizu Wang, Karen Ke Lin

**Affiliations:** 1Institute of Materials Research and Engineering, Agency for Science, Technology and Research, 2 Fusionopolis Way, Singapore 138634, Singapore; eugene_soh@imre.a-star.edu.sg (E.S.J.H.); n-zhang@imre.a-star.edu.sg (N.Z.); zhuq@imre.a-star.edu.sg (Q.Z.); 2Institution of Sustainability for Chemical, Energy and Environment (ISCE2), 1 Pesek Road, Jurong Island, Singapore 627833, Singapore; 3School of Chemistry, Chemical Engineering and Biotechnology, Nanyang Technological University, 21 Nanyang Link, Singapore 637371, Singapore

**Keywords:** THz spectral, eye drop, water content, graphene oxide, CNT

## Abstract

With chronic ocular diseases such as glaucoma and dry eye syndrome, patients have to apply eye drops over the long term. The therapeutic effects of eye drops depend on the amount of drug contained and the stability of the solution. In addition, contamination during usage and transport can also negatively affect the quality and efficacy of eye drops. The current techniques for the characterization of eye drops are often complicated and time-consuming. Developing a fast and non-invasive way of accurately measuring eye drop quality remains an ongoing challenge. The biggest challenge and primary prerequisite for the application of this new detection technique for eye drops is the obtention of a sufficient spectral response and resolvable signal, considering the large background signal contributed by water. In this work, we use terahertz (THz) attenuated total reflection (ATR) spectroscopy combined with a sensitive hybrid graphene oxide (GO) and carbon nanotube (CNT) thin-film sensors to obtain distinct THz spectral signals in commercial eye drops. Various commercial eye drop products have been tested, and we show that they can be differentiated via their spectral signals. Our results provide a solid foundation for the future fine analysis of eye drops and the detection of their quality. Furthermore, THz spectroscopy combined with GO/CNT films has significant potential and advantages for the non-destructive characterization of aqueous pharmaceutical products.

## 1. Introduction

Many environmental factors, such as ambient temperature, humidity, atmospheric oxygen, and light can affect the stability of pharmaceutical products [1]. Hence, non-destructive testing is crucial for evaluating the effects of environmental factors on the performance and stability of a formulated product, and will help manufacturers to suggest appropriate labels with a prediction of the product’s shelf life and proper storage conditions. Inappropriate storage conditions can lead to the contamination of eye drops designed for multiple use, with the existing technology detecting contamination levels of up to 38% [2,3,4,5,6,7,8]. Any physical, chemical, or microbiological changes in the product can negatively impact the patient’s eye health [9,10,11].

To characterize and monitor product stability, common methods include UV–Vis spectrophotometry [12], high-performance liquid chromatography [13], and mass spectrometry [14]. However, there are some limitations to these techniques. UV–Vis spectroscopy only probes electronic transitions and does not provide further information such as the molecular vibrations and rotations of the sample. Liquid chromatography and mass spectrometry do not provide information on the spectral absorption and transmission properties of the sample. Even though they are widely used in material characterization, there are still sampling limitations to these methods. Raman spectroscopy, usually using visible-to-near-infra-red light, could overcome some of these limitations, but the signal tends to be weak, and it becomes difficult to measure low-concentration samples. This, however, can be improved using a surface-enhanced Raman technique based on gold nanoparticles to increase the detection sensitivity; however, this enhancement is limited. Compared with Raman spectroscopy, terahertz spectroscopy offers several advantages for the molecular measurement of materials, such as non-ionizing radiation, a high penetration depth, a higher sensitivity to water absorption, and organic molecular material identification [15,16].

In this work, both the transmission and pulsed terahertz attenuated total reflectance spectroscopy (ATR) modes of THz spectroscopy were explored. The transmission mode provides reliable parameters and extraction results; however, for high-hydration eye drop samples, the strong water absorption means that the transmission mode is difficult to apply. In the THz ATR mode, the sample material is placed directly onto a high-refractive-index prism [17]. The THz incident beam, at certain angles, will experience total internal reflection in the prism interface, producing an evanescent field that attenuates exponentially inside the sample. Utilizing the evanescent wave generated between the sample and crystal, the sample’s spectral properties are measured via the detection and analysis of the reflected THz beam [18]. The ATR mode has the advantage of being very sensitive to interface changes and provides reliable results for parameter extraction [19]. Furthermore, due to the evanescent field at the interface, ATR has a higher sensitivity and is more appropriate for aqueous and gas sample characterization.

Thin-film surface modification is one way to modify the surface reflectance [20]. Graphene-based nanomaterials have attracted considerable attention in the field of terahertz spectroscopy due to their unique properties and their potential to enhance the performance of terahertz devices. Graphene is a single layer of carbon atoms arranged in a two-dimensional honeycomb lattice, and has excellent electrical, optical, and mechanical properties. These properties make graphene an ideal candidate for enhancing terahertz spectroscopy in several ways, including an increased sensitivity, broadband absorption, ultrafast response, and tunability. However, pristine graphene is expensive and difficult to process.

Graphene oxide (GO) offers several advantages over pristine graphene, making it an attractive alternative for certain applications. Graphene oxide has a lower cost compared to pristine graphene. GO can be produced through a relatively simpler and more cost-effective synthesis process. This makes GO a more economically viable option for large-scale production and commercial applications. GO includes oxygen-containing functional groups, such as hydroxyl, carboxyl, and epoxy groups, on its surface. These functional groups mean that GO is highly dispersible in various solvents, including water. The ease of dispersion and processing of GO allows its incorporation into different matrices or the fabrication of thin films, coatings, and composite materials. The presence of oxygen-containing functional groups on the surface of GO offers an improved solubility in water and other polar solvents. This solubility enables easier handling of GO and its integration into various systems. Additionally, GO is generally more stable than pristine graphene due to the presence of these functional groups, which provide a better resistance to aggregation and improve its overall stability in a solution. Moreover, the presence of functional groups on the GO surface offers the opportunity for further chemical modifications and functionalization.

In this work, substrates coated with graphene oxide (GO), carbon nanotubes (CNTs), and a combination of both are used as plasmonic enhanced sensors for eye drop detection. We demonstrate how the THz ATR mode combined with the GO and GO + CNT sensors enhanced the eye drop spectral responses.

## 2. Experimental Setup

### 2.1. Instrumentation: THz Transmission and ATR Spectroscopy

Figure 1a shows the THz transmission schematic setup. Femtosecond fiber lasers with a 250 MHz repetition rate and 90 femtosecond laser pulses at a 1.56 μm central wavelength are directed onto two photoconductive antennas used as a fiber-coupled emitter and receiver. The THz pulse is generated with a bandwidth coverage up to 3 THz. The sample is placed on a translation stage, which allows for the precise positioning of the sample relative to the THz beam. The THz beam can be focused onto the sample using a THz TPX lens. For the THz transmission method, the eye drops are placed in a quartz bottle between the THz source and the detector. A reference signal is taken without the sample present, and the transmission spectrum of the sample can be analyzed to obtain the sample’s THz properties, such as its refractive index and absorption coefficient.

As shown in Figure 1b, for the attenuated total reflection (ATR), the THz beam from the emitter enters the high-refractive-index crystal (i.e., high-resistivity Si) through a 500 μm thick silicon substrate (either plain Si or functionalized with CNT, GO, or CNT+GO thin films), into the sample, where it undergoes reflection or absorption. It is known that 500 μm of Si is thin enough that it is transparent to the THz evanescent wave. A glass cover slip is placed on top of the eye drop to prevent evaporation during the experiment. The reflected THz radiation then propagates back through the crystal and is detected by the THz detector. For each measurement, we take the THz spectrum in one spot, integrated from about 10 to 20 secs. The spot size is from about 2 to 3 mm in diameter, and the signal acquired represents the average signal of the region given by the spot size. The ATR setup has advantages over traditional transmission measurements in that it can be used to study very thin films or liquid samples. Additionally, the ATR setup is relatively easy to use and does not require any sample preparation, making it a popular choice for application in THz spectroscopy.

### 2.2. Sensor Preparation

The thin-film-coated Si is developed to enhance the surface plasmonic effect during measurement. To prepare the GO-based sensor, graphene oxide (graphene; S.A. dispersibility; polar solvents; PH: 1.8–2; concentration: 2.5 wt%) is spin-coated on a high-resistivity Si substrate and annealed at 120 °C for 10 min. Similarly, CNT and GO+CNT substrates are prepared using the same spin-coating process after the mixing in of water or methanol. The concentration of CNT and GO can be adjusted to optimize the sensing performance. We vary the concentration of CNT from 20% to 50% by weight, with 50 wt% giving the highest sensitivity. Then, we obtain the three types of nano-sensors based on the dried GO, CNT, and GO+CNT thin films. The commercial eye drops were drop-casted onto the thin-film surface, which is in contact with the prism, to detect the THz spectrum response. Figure 1c shows the schematic illustration of the GO and GO+CNT thin films formed on top of Si and used in the THz ATR mode for eye drop sensing.

### 2.3. The THz ATR Theory Calculation

The absorption spectrum of the sample and refractive index will give an insight into the chemical composition and molecular structure of the material. We first obtain the terahertz ATR spectrum (signal intensity vs. time) using a terahertz time-domain spectroscopy system, as shown in Figure 1. Subsequently, by applying a fast Fourier transform, we obtain the amplitude and phase of the electric field in the frequency domain. The attenuated total reflectivity *R*(ω) and the phase spectrum $\Phi \left(\omega \right)$ in the frequency domain can be determined from:(1)$$R\left(\omega \right)={\left|\frac{r_{12}\left(\omega \right)}{r_{ref}\left(\omega \right)}\right|}^{2}$$
(2)$$\Phi \left(\omega \right)=Arg\left[\frac{r_{12}\left(\omega \right)}{r_{ref}\left(\omega \right)}\right]$$
where $r_{12}\left(\omega \right)$ and $r_{ref}\left(\omega \right)$ are the Fresnel’s reflection coefficients of the prism–sample interface and the prism–air interface, respectively. The Fresnel reflection coefficient, $r_{12}\left(\omega \right)$, is given by:(3)$$r_{12}\left(\omega \right)=\frac{n_{1}\{\omega )\sqrt{(1-{\left(\frac{n_{1\;}\left(\omega \right)}{n_{2}\left(\omega \right)}\right)}^{2}\sin^{2}\theta }-{\tilde{n}}_{2}\left(\omega \right)cos\theta }{n_{1}\{\omega )\sqrt{(1-{\left(\frac{n_{1\;}\left(\omega \right)}{n_{2}\left(\omega \right)}\right)}^{2}\sin^{2}\theta }+{\tilde{n}}_{2}\left(\omega \right)cos\theta }$$
where *θ* is the incident angle, and *n*_1_*(ω)* is the complex refractive index of the ATR prism. The complex refractive index of the sample, ${\tilde{n}}_{2}=n\left(\omega \right)-ik\left(\omega \right)$, where *n* and *k* are the refractive index and the extinction coefficient, respectively, can be calculated from Equations (1)–(3). From these, we can calculate the refractive index and the absorption coefficient at the frequency domain.

## 3. Results and Discussion

**1.** 
**Eye drop response in THz transmittance and ATR method**


To investigate and compare the effect of the signal strength and sensitivity of the detection modes for eye drops via THz spectra, we characterized the eye drop on the Si sensor in both the transmittance and ATR modes. Figure 2a shows the absorption coefficient and refractive index spectral obtained using the transmission mode. There are several absorption peaks, at 0.2 THz, 1.7 THz, and 2.4 THz, demonstrated in Figure 2a. The refractive index spectra in Figure 2b show high values of about 3.5 because there is a large amount of water in the eye drop solution. There are peak shifts, i.e., at 0.2 THz and 2.4 THz, and intensity changes, though they are not obvious. There are differences at lower frequencies for Rohto eye drops and DI water.

Due to the high absorption of water in the THz range, it is not practical to use the transmission mode to measure aqueous types of samples, such as eye drops. ATR-mode THz spectroscopy offers several advantages over transmission-based THz spectroscopy, such as requiring minimal sample preparation. Moreover, for transmission measurements, the sample needs to be thin enough to allow the THz radiation to pass through. This can be challenging and can require additional sample preparation techniques. THz ATR, on the other hand, allows for the direct analysis of solid or liquid samples without the need for thinning or special sample-handling.

Figure 3 shows the absorption coefficient and refractive index spectra for both DI water and Rohto eye drops on a silicon substrate using THz ATR mode. There is a greater difference in signal between the water and Rohto eye drops (i.e., indicated by the difference in intensities throughout the frequency range from 0 to 2.0 THz). However, neither the DI water nor the Rohto eye drops demonstrated obvious absorption peaks. THz ATR spectroscopy could enhance the sensitivity. When the THz radiation interacts with the sample at the interface, it penetrates a small depth of the sample. This penetration depth allows for increased interaction between the THz radiation and the sample, leading to improved sensitivity for thin layers or surface-related phenomena. To further enhance the signal sensitivity, we prepare Si substrate modified with CNT, GO, and a mixture of GO+CNT and use ATR mode to study the plasmonic enhancement effect of the substrate and the eye drop detection.

**2.** 
**Eye drop response on GO- and CNT-coated Sensors**


Figure 4 shows the SEM images of a GO thin-film coating on a Si substrate. Large sheets of GO are used for preparing the sensor, and the GO films are observed to be continuous and uniform. However, at the sheet boundaries, overlaps occur, and wrinkles of a few tens of nanometers appear. These wrinkles and boundaries could have significantly different physical and chemical properties. This overall flat nanostructure can effectively increase the plasmonic effect on the surface. Furthermore, the XPS spectra also verify the presence of a C-O band at 286.6 eV.

CNT, GO, and GO+CNT are dispersed in DI water or methanol and spin-coated onto the silicon substrates. Figure 5a shows the optical microscopy images of the CNT, GO, and hybrid-GO+CNT-mixture thin films. Compared to CNT, the GO+CNT hybrid mixture is dispersed more uniformly. As our aim is to verify the utility of GO, CNT, or GO+CNT films, and to improve the THz detection and sensing performance, we choose three sensors with which to perform proof-of-concept experiments (as shown in the figures below).

Firstly, we need to calibrate or obtain the fundamental lines of these thin films in the THz spectrum. We study the effect of GO, CNT, and GO+CNT on the silicon substrate by spin-coating these solutions onto silicon. Figure 5c and d show the different thin films (i.e., GO, CNT, and GO+CNT hybrid) on a Si substrate, and their time domain and frequency domain in the ATR configuration. Si/GO shows an increased reflection intensity compared to a bare Si substrate. We observed an even greater enhancement in reflection THz intensity after coating the CNT and GO+CNT thin films. The presence of the GO layer can support the excitation and propagation of surface plasmons, which are collective electron oscillations at the GO surface. The interaction between the incident THz radiation and these surface plasmons can result in enhanced field confinement, leading to stronger THz signal. The GO+CNT hybrid film shows the highest reflection and lowest absorbance from 1 THz to 3 THz. We observe that Si modified with different thin films shows distinct differences in the THz spectrum. It is evident that the plasmonic effect of the interface has a significant impact on the THz spectrum. The improved reflection can be beneficial in the sensitive detection or characterization of the sample, as it improves the overall signal strength and enhances the sensitivity of the measurement.

This enhancement can be attributed to synergistic interface plasmonic effects: both graphene and CNTs exhibit unique plasmonic properties in the THz frequency range. When combined, they can exhibit synergistic effects that enhance the interaction with incident THz radiation. The graphene layer and CNTs act as complementary plasmonic components, contributing to increased field confinement, absorption, and reflection. Compared to bare Si and Si/GO, when CNT is added (in the case of Si/CNT and Si/GO+CNT), the absorption intensities for both decreased significantly. This is due to the increased conductivity of CNT, which results in increased electrical shielding and, hence, attenuates the interaction between the eye drops and the thin film. This suggests that too high a conductivity can, instead, affect the absorption of terahertz light at the interface, thus weakening the sensitivity of the signal. Moreover, the thickness of the coated thin film can affect the sensitivity. A thick layer would reduce the optical penetration depth and, hence, lower the sensitivity, while too thin a layer could shift the plasmonic frequencies, causing a mismatch with the THz frequency range of interest. Extremely thin layers may result in increased absorption losses. The thickness of the layer can affect interference patterns and the overall signal. Therefore, it is important to optimize the concentration and thickness of the CNT+GO layer to maximize the peak signal contrast. In this work, we found a film thickness of 100 nm to be suitable. By controlling the concentration of the GO solution (0.4–2wt %) and the speed of spin-coating, we obtained GO thin films of different thicknesses from 30 to 100 nm. Finally, we selected the 30 nm film thickness for GO, CNT, and GO+CNT thin films. Its morphology is shown in Figure 4. In the reflection spectrum, more peak information is demonstrated, which is likely to be valuable in the analysis of the composition in the future.

**3.** 
**The response of the THz spectra of the eye drops on nano thin-film sensors**


Figure 6 shows the THz time domain spectra of the Rohto eye drops on the different sensors. The first time domain peak is due to the prism and Si substrate, while the second interface, modified Si sensor, and eye drop contribute to the second peak in the time domain. The schematic of the experimental setup is shown in Figure 6b, and Figure 6c shows the enlarged second peaks. Compared with the unmodified Si, when the Rohto eye drop is applied, significant absorption can be observed for all the sensors. The bare Si and the Si with the Rohto eye drop show the least change, which proves that the incident THz beam has the least interaction with the eye drop on the bare Si substrate. When the eye drop is deposited on the Si/CNT thin film, the signal is greatly diminished. One possible reason could be that the CNT film is not continuous, as seen in Figure 5a. Another possible reason could be the poor adhesion of the CNT thin film to the Si substrate. This could cause the CNT film to be washed away when the eye drop is placed above it. Si/GO and Si/GO+CNT demonstrated the largest change. As the previous results were based on GO and GO/CNT on Si substates, the incorporation of GO and CNTs can introduce additional electric field confinement near the interface. This electric field confinement results in increased THz reflection without eye drops. Figure 7 shows the absorption and refractive index spectra of the Rohto eye drop on the Si sensors modified with CNT, GO, and GO+CNT as a function of the frequency. Strong absorption interactions in the eye drops are achieved on GO and GO+CNT thin films due to the plasmonic effects of the thin films. On Si/GO+CNT, the Rohto eye drops have an even higher absorption. GO together with CNTs exhibits a high carrier mobility, which results in an increased conductivity and charge carrier density at the interface. This enhanced conductivity facilitates the efficient transport of charge carriers, leading to even more improved plasmonic effects.

Without GO thin film, the DI water and Rohto are difficult to differentiate (see Figure 3a and Figure 8a). With GO thin film, the high carrier mobility and electrical conductivity enable the efficient detection and amplification of THz signals. When incorporated into THz detectors, GO-based devices can exhibit a high responsivity and sensitivity, enabling the detection of weak THz signals with improved signal-to-noise ratios. Furthermore, GO offers remarkable broadband absorption characteristics, enabling it to absorb a wide range of THz frequencies. We compare the spectral response of the absorption efficiency and the refractive index of the Rohto eye drops on the different substrates. We can visualize that the continuous GO and GO+CNT thin-film-coated detectors have stronger absorption efficiencies at 0.5–2.5 THz, indicating a more pronounced surface plasmonic effect. The change in refractive index is also more pronounced at 0–2 THz. By integrating GO and CNTs into THz absorbers, researchers can design structures with tailored absorption properties, leading to an improved sensitivity and selectivity in THz spectroscopy.

Moreover, we show that our sensor can be applied to help us distinguish between a range of different types of eye drops. Figure 8 shows the THz absorption coefficient and refractive index comparison for a series of commercial eye drops, including Optrex, Rohto, and Systane. The results demonstrated clear differentiation between each product. We note that, although Optrex and Systane exhibit a similar absorption coefficient at the low and high frequencies (i.e., <0.5 THz and >1.7 THz), they exhibit a distinguishable absorption coefficient between 0.5 and 1.6 THz. THz ATR techniques with interface-enhanced plasmonic sensing have the potential to be applied to eye drop characterization or even quality and composition monitoring. With the enhancement provided by the GO/CNT surface functionalization, slight shifts in the peaks can be detected, which could indicate contamination of the order of ppm. It is important to note that the specific enhancement achieved via the combination of graphene and CNTs in THz ATR setups will depend on factors such as the composition, arrangement, and concentration of the materials, as well as the characteristics of the incident THz radiation. Experimental investigation and optimization are typically performed to maximize the benefits of the graphene–CNT composite for enhanced THz ATR reflection.

## 4. Conclusions

A comparative test of Rohto eye drops and deionized water was carried out using the ATR mode of terahertz. By optimizing the Si substrate with a thin film of CNT, GO, or GO+CNT, we show that GO+CNT achieves the optimum contrast. A high contrast is required when spectroscopically testing eye drops for degradation and contaminants. The absorption contrast can be increased via the appropriate surface modification of thin films, making ATR an attractive method for the reliable, rapid, and non-destructive testing of eye drops and other pharmaceutical products. Moreover, coating substrates with carbon thin films (i.e., GO, CNT, and GO+CNT on Si) can improve their sensitivity to water and organic molecules due to enhanced surface plasmon excitation. Among them, the GO+CNT-coated Si sensor achieves the best absorption and resonance signals. Furthermore, we show that the thin film coatings enable us to differentiate between several brands of eye drops. Our results demonstrate the effectiveness and versatility of the thin-film-functionalized sensors to differentiate between different brands of eye drops. We provide the feasibility of a rapid, non-destructive assay that paves the way for future research into aqueous eye drops and aqueous pharmaceuticals.

## Figures and Tables

**Figure 1 sensors-23-08290-f001:**
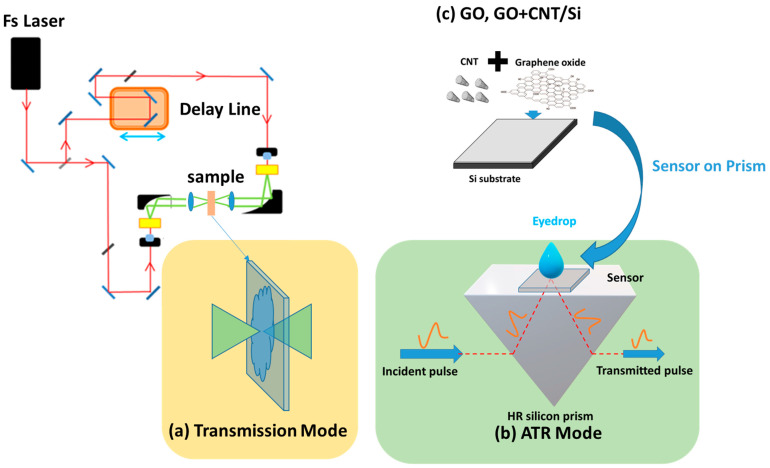
Diagram of (**a**) the transmission mode and (**b**) the ATR mode for eye drop measurement. (**c**) GO, CNT, and GO+CNT thin films on a Si substrate for THz ATR sensing.

**Figure 2 sensors-23-08290-f002:**
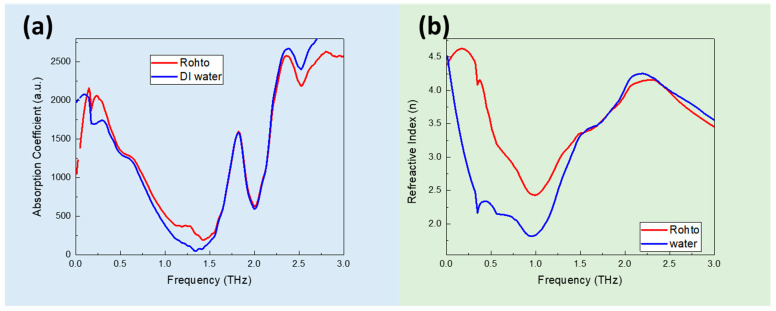
(**a**) The absorption coefficient spectra and (**b**) the refractive index of DI water and Rohto eye drops using the THz transmittance mode.

**Figure 3 sensors-23-08290-f003:**
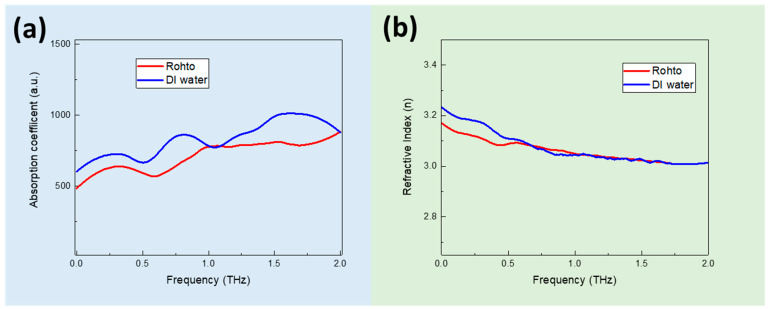
The THz absorption coefficient (**a**) and the refractive index spectra (**b**) for DI water and Rohto eye drops on a Si substrate using THz ATR mode.

**Figure 4 sensors-23-08290-f004:**
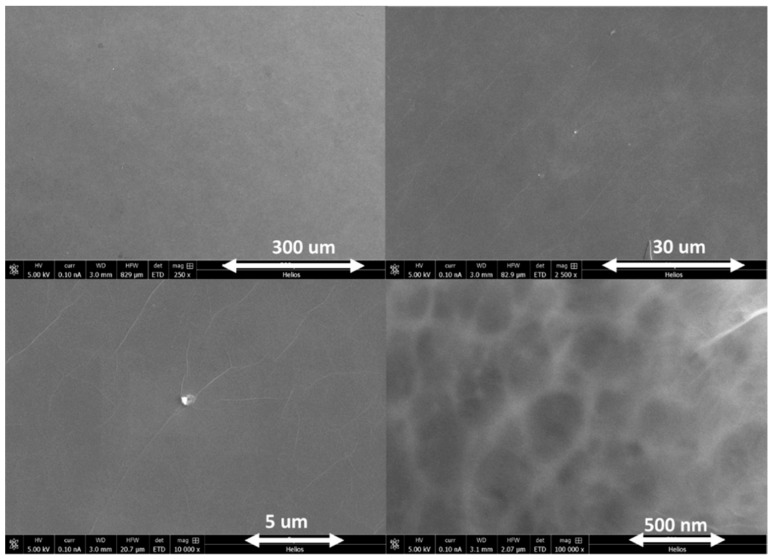
SEM images of GO film coatings on a Si substrate at different scales. We show that the synthesized GO films appear uniform and continuous across length scales from 500 nm to 300 µm.

**Figure 5 sensors-23-08290-f005:**
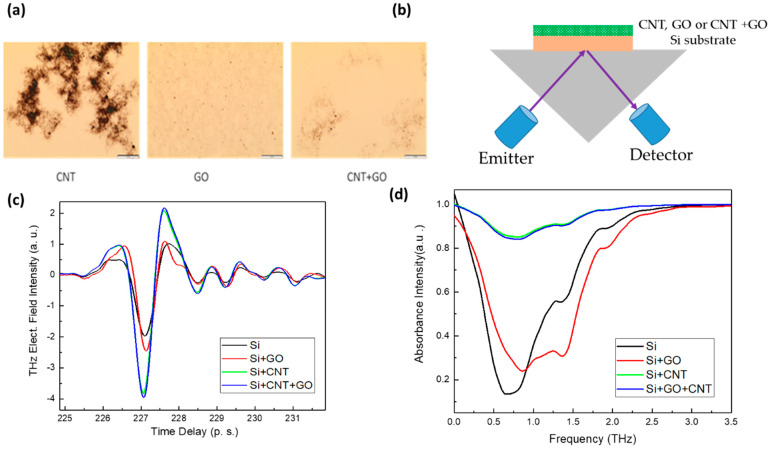
(**a**) Optical images of the CNT, GO, and GO+CNT thin films on a Si substrate, (**b**) schematic of the nanosensor modified ATR prism interface, (**c**) THz time-domain spectroscopy, and (**d**) the absorbance intensity with the THz frequency.

**Figure 6 sensors-23-08290-f006:**
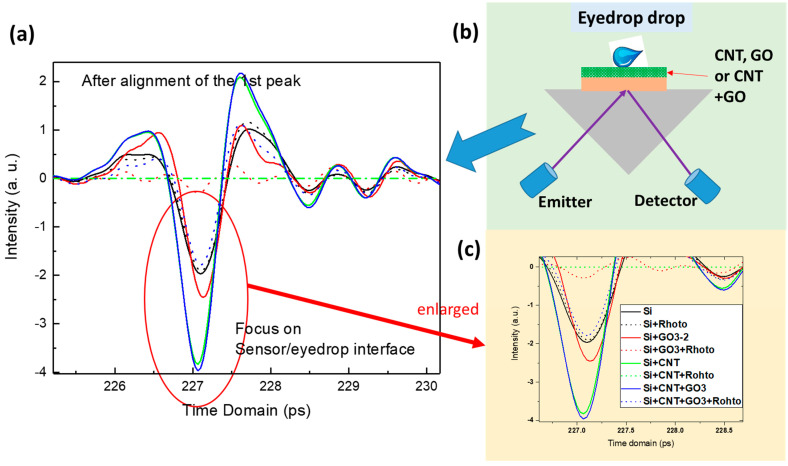
(**a**) The THz time domain spectra of the Rohto eye drop on the different sensors, (**b**) schematic of the experimental setup with the eye drop added, and (**c**) the enlarged second peaks.

**Figure 7 sensors-23-08290-f007:**
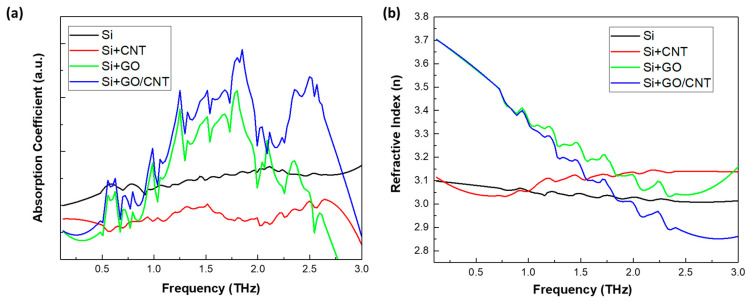
(**a**) The absorption coefficient of the Rohto eye drop on different thin-film sensors, and (**b**) the refractive index.

**Figure 8 sensors-23-08290-f008:**
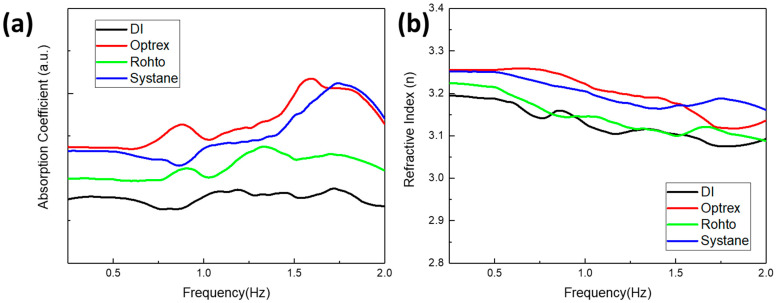
(**a**) The absorption coefficients of the three eye drops (Optrex, Rohto, and Systane) and DI water, and (**b**) the refractive index on the GO-modified Si sensor.

## Data Availability

The data presented in this study are available on request from the corresponding author. The data are not publicly available due to A*Star data protection policy.

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
