# Peer review of "Terahertz Attenuated Total Reflection Spectral Response and Signal Enhancement via Plasmonic Enhanced Sensor for Eye Drop Detection"

_sensors, 2023, doi:10.3390/s23198290_

Round 1

Reviewer 1 Report (New Reviewer)

This manuscript is devoted to the development of a fast and non-invasive method for accurately measuring the quality of eye drops using terahertz attenuated total reflection spectroscopy in combination with a sensitive graphene oxide/carbon nanotube hybrid thin film. Various commercial eye drops have been tested using this technique. The authors believe that the results provide a solid basis for future accurate analysis and detection of eye drops. The manuscript can be recommended for publication in Sensors, after taking into account one important and several minor issues.

Important issue. 1. In the conclusion it is written: "... we show that thin-film coatings make it possible to distinguish between several brands of eye drops." It is not very clear on what basis such a conclusion was made. The absorption coefficients of the two eye drops (Optrex and Systane) are very close to each other, see Figure 8 (a).

Minor issues.

1. Lines 50 - 52. The meaning of the sentence "Compared to Raman spectroscopy, terahertz spectroscopy offers a number of advantages for molecular measurements of materials" is not clear. Raman spectroscopy in the terahertz region is widely used in the study of organic semiconductors, see https://doi.org/10.1021/acs.cgd.3c00634 and references therein.

2. Line 111. The abbreviation "um" needs to be defined.

3. Line 142. The title of the section is extremely unfortunate.

4. Line 168. The meaning of the term "frequency spectrum" is unclear.

5. "Absorption (a.u.)", "absorption coefficient (cm-1)" are used on the Y-axis of figures 2 (a) and 3 (a), respectively. One unit of measure must be used.

Author Response

This manuscript is devoted to the development of a fast and non-invasive method for accurately measuring the quality of eye drops using terahertz attenuated total reflection spectroscopy in combination with a sensitive graphene oxide/carbon nanotube hybrid thin film. Various commercial eye drops have been tested using this technique. The authors believe that the results provide a solid basis for future accurate analysis and detection of eye drops. The manuscript can be recommended for publication in Sensors, after taking into account one important and several minor issues.

Important issue. 1. In the conclusion it is written: "... we show that thin-film coatings make it possible to distinguish between several brands of eye drops." It is not very clear on what basis such a conclusion was made. The absorption coefficients of the two eye drops (Optrex and Systane) are very close to each other, see Figure 8 (a).

Reply: We appreciate the comments made by the review, and we have found significant spectral and response differences between commercial eye drops and deionized water through multiple measurements. We also found that the three eye drops have different absorption and refractive index responses near some terahertz frequencies. This is likely to come from their different chemical compositions. Optrex and Systane may have similar absorption coefficients at the low and high frequencies (<0.5 THz, >1.6THz), but they have more contrast and are distinguishable at other frequencies such as (~0.8THz and 1.6THz). We have included this discussion in line 217-218.

Minor issues.

  1. Lines 50 - 52. The meaning of the sentence "Compared to Raman spectroscopy, terahertz spectroscopy offers a number of advantages for molecular measurements of materials" is not clear. Raman spectroscopy in the terahertz region is widely used in the study of organic semiconductors, see https://doi.org/10.1021/acs.cgd.3c00634 and references therein.

Reply: Yes, we revised this sentence and cited this reference.

  1. Line 111. The abbreviation "um" needs to be defined.

Reply: We have corrected it to μm.

  1. Line 142. The title of the section is extremely unfortunate.

Reply: Yes, we revised to the results on GO and CNT coated sensor.

  1. Line 168. The meaning of the term "frequency spectrum" is unclear.

Reply: I think reviewer point the line 156, we revised to frequency domain.

  1. "Absorption (a.u.)", "absorption coefficient (cm-1)" are used on the Y-axis of figures 2 (a) and 3 (a), respectively. One unit of measure must be used.

Reply: we revised to a.u.

Reviewer 2 Report (New Reviewer)

The paper titled “THz ATR Spectral Response and Signal Enhancement by Plasmonic Enhancement Sensor for Eye Drop Detection”, presents a novel non-destructive method for evaluation of eye drops through terahertz attenuated total reflection.

Comment 1:

The manuscript is relevant to the field and could lead to further research towards non-destructive analysis of eye drops.

Comment 2:

The manuscript addresses important need in the field which may have been overlooked.

Comment 3:

The figures are presented well. Figure 6 can be improved.

The experiments performed are adequately elaborated.

The results presented are clear and in line with the conclusions arrived.

Author Response

Reviewer 2

The paper titled “THz ATR Spectral Response and Signal Enhancement by Plasmonic Enhancement Sensor for Eye Drop Detection”, presents a novel non-destructive method for evaluation of eye drops through terahertz attenuated total reflection.

Comment 1:

The manuscript is relevant to the field and could lead to further research towards non-destructive analysis of eye drops.

Reply: Thanks

Comment 2:

The manuscript addresses important need in the field which may have been overlooked.

Reply: Thanks

Comment 3:

The figures are presented well. Figure 6 can be improved.

Reply: Yes, we re alignment the figure 6.

The experiments performed are adequately elaborated.

The results presented are clear and in line with the conclusions arrived.

Reply: Thanks

Reviewer 3 Report (Previous Reviewer 1)

The figure quality needs to be further improved, like Fig. 6. In addition, the authors need to check the data accuracy of absorption coefficient (AC) in Figs. 7 and 8, where the AC of the samples at 1 THz exceeds 500, even 1000 cm^-1. This is quite unreasonable.  

Author Response

Reviewer 3

The figure quality needs to be further improved, like Fig. 6. In addition, the authors need to check the data accuracy of absorption coefficient (AC) in Figs. 7 and 8, where the AC of the samples at 1 THz exceeds 500, even 1000 cm^-1. This is quite unreasonable.  

Reply: We have improved the resolution and formatting of Figure 6, and we thank the reviewers for pointing out errors in the graphing of our data. We have changed the units to a.u. since we calibrated all spectra against the first peak position as a reference. Here we have standardized the absorption ratio and base calibration for all the plots.

Reviewer 4 Report (New Reviewer)

In this work, the authors present a THz ATR spectral response for eye drop detection. This work is interesting and could be useful for the droplet detection. However, the structure and writing of the paper need to be improved. Some major revions are needed, as follows.

1. Abbreviations should not appear in titles, such as Thz and ATR. 

2. Only 20 references were presented in this work, and all of them appeared in the introduction part. The literature review part must be improved.

3. Figure 2(a): there is a mistake in x-axis.

4. Figure 4: what are the differences of these four SEM images, it should be described in the captions.

5. How dose the sensor work? What are the parameters for the sensor? The author should present the performance of the sensor.

Author Response

Reviewer 3

In this work, the authors present a THz ATR spectral response for eye drop detection. This work is interesting and could be useful for the droplet detection. However, the structure and writing of the paper need to be improved. Some major revions are needed, as follows.

  1. Abbreviations should not appear in titles, such as Thz and ATR. 

Reply: Yes, we revised the title to “Terahertz Attenuated Total Reflection Spectral Response and Signal Enhancement by Plasmonic Enhanced Sensor for Eye Drop Detection”

  1. Only 20 references were presented in this work, and all of them appeared in the introduction part. The literature review part must be improved.

Reply: Yes, we added the citation at the suggestion of other reviewers. Due to the limited research in this area, the literature available to us was limited.

  1. Figure 2(a): there is a mistake in x-axis.

Reply: Yes, we revised figure 2 and figure 3.

  1. Figure 4: what are the differences of these four SEM images, it should be described in the captions.

Reply: Many literature reports that GO coated nanofilms are non-continuous lamellar structures, which is not favorable for making probe responses as optical sensors. Our own synthesized and coated GO films appears uniform and continuous from 300 µm to 500nm range. Hence, there is no excess scattering and diffraction of THz light.

  1. How dose the sensor work? What are the parameters for the sensor? The author should present the performance of the sensor.

Reply: The sensor thickness and composition of GO film are selected to maximise plasmonic enhancement of the THz signal, thus we are able to differentiate between the different eyedrops. We have chosen a thickness of 30nm GO film and concentration of GO between 0.4-2%wt. Compare with the background and eyedrop absorption coefficient and reflex index change at selected peak, we can get different sensitivity of eyedrop on sensor. For example: Compared with Figure 3 and 8, we observed the density changing of absorption coefficient and refractive index increase 5 and 3 times at 1.5THz.

Round 2

Reviewer 4 Report (New Reviewer)

Accept in present form.

Author Response

Comments and Suggestions for Authors

Accept in present form.

Reply: Thanks

This manuscript is a resubmission of an earlier submission. The following is a list of the peer review reports and author responses from that submission.

Round 1

Reviewer 1 Report

The authors proposed a nanosensor based on GO and CNT to detect eye drop and enhance the detection sensitivity through THz ATR measurements. The design of the nanosensor on the ATR prism surface is novel and the results are interesting. This work is worth to be published after these issues are addressed:

(1)    From the description in lines 93-95, it seems there is no Si substrate between the prism surface and the liquid sample, but in Figs. 1 and 4, the Si substrate does exist. Please make it clear. Furthermore, if the Si substrate exists, what is its thickness? This is very important since the evanescent wave cannot penetrate too thick.

(2)    Fig. 5 is not formal in an article. Please revise.

(3)    In Fig. 2, why the curves for the Si substrate in top and bottom figures are not consistent? Actually, the top and bottom figures should be combined.

(4)    What is GO3? What is the difference between GO and GO3?

(5)    The thickness of the GO or CNT film has important effect on the absorption of the detected eye drop. The authors need to add the data/plot to discussion.

(6)    The explanation in lines 204-225 is not clear. There are plasma effect and conductivity shielding effect when the Si substrate is coved by GO or CNT and GO/CNT. The eventual effect will depend the ratio and the relative thickness of the GO or CNT in the film. The authors need to give a clearer discussion.

(7)    Why are there 8 curves for comparison in Fig. 5 but 4 curves in Fig. 6?       

Author Response

The authors proposed a nanosensor based on GO and CNT to detect eye drop and enhance the detection sensitivity through THz ATR measurements. The design of the nanosensor on the ATR prism surface is novel and the results are interesting. This work is worth to be published after these issues are addressed:

(1)    From the description in lines 93-95, it seems there is no Si substrate between the prism surface and the liquid sample, but in Figs. 1 and 4, the Si substrate does exist. Please make it clear. Furthermore, if the Si substrate exists, what is its thickness? This is very important since the evanescent wave cannot penetrate too thick.

Ans: Thanks remind us, have revised in text page 2 and figures(1,3,4), highlight in yellow.

(2)    Fig. 5 is not formal in an article. Please revise.

Ans: Yes, revised in page 4, highlight in yellow.

(3)    In Fig. 2, why the curves for the Si substrate in top and bottom figures are not consistent? Actually, the top and bottom figures should be combined.

Ans: Fig. 2 is replotted for better understanding.

(4)    What is GO3? What is the difference between GO and GO3?

Ans: Yes, this has been corrected and all change to GO.

(5)    The thickness of the GO or CNT film has important effect on the absorption of the detected eye drop. The authors need to add the data/plot to discussion.

Ans: High resistivity Si substrate 400um. There is a kind of refractive index solution in between the Si substrate and Si prism to reduce the loss of the THz wave. THz wave could penetrate through the high resistivity Si substrate and reach the upper interface of eyedrop/sensor.

(6)    The explanation in lines 204-225 is not clear. There are plasma effect and conductivity shielding effect when the Si substrate is coved by GO or CNT and GO/CNT. The eventual effect will depend the ratio and the relative thickness of the GO or CNT in the film. The authors need to give a clearer discussion.

Ans: Yes, we redescribe and discuss this section. Changes and additions have been inserted into the text(line 209-214, highlight in yellow)

(7)    Why are there 8 curves for comparison in Fig. 5 but 4 curves in Fig. 6?       

Ans:  The number and order of the graphics we have rearranged is more specific to the requirements of the review. The time test spectra for all eight test samples are given in Figure 7. In contrast, Figures 6, 8 are separate graphs giving comparisons of the groupings under different tests, which are more amenable to analysis and discussion.

Reviewer 2 Report

The reviewer thinks the topic of this manuscript is suitable to be published in Sensors. The topic of this paper is the study and optimization of a highly sensitive hybrid graphene oxide and carbon nanotube sensor which obtained the first clear and highly sensitive THz spectral signal in a commercial eye drop. This paper is well organized and the results are meaningful for future fine analysis and detection of aqueous pharmaceuticals. However, before publication, there are also some suggestions for the authors to make moderate revisions.

Materials and Methods

1) Page 3 Line 109.

Perhaps you can explain here why can the signal represent the average effects from the whole samples?

2) Page 3 Line 116.

I wonder if methanol would affect the experimental results here, and could you explain why methanol is added? What are the benefits of doing so?

3) Page 4 Figure 1(b).

The two curves in the graph have a break around 0.1 THz, and for the black curve, there is a missing data portion of absorbance after approximately 2.6 THz. It is recommended to redraw the graph.

4) Page 5 Figure 2.

Here I suggest that Figure 2 should add a legend to clearly differentiate between different curves.

5) Page 6 Figure 3.

The four images here do not correspond to the subsequent text descriptions. I suggest labeling each of these four images separately to facilitate clearer descriptions in the following text.

6) Page 8 Line 223.

Could you provide a more detailed description of how the detector thickness affects the signal and explain the mechanisms behind it?

About the figure issues:

I have noticed that the figures in your paper are not very clear. I suggest exporting them as vector graphics for improving the quality of them.

Author Response

Review 2

he reviewer thinks the topic of this manuscript is suitable to be published in Sensors. The topic of this paper is the study and optimization of a highly sensitive hybrid graphene oxide and carbon nanotube sensor which obtained the first clear and highly sensitive THz spectral signal in a commercial eye drop. This paper is well organized and the results are meaningful for future fine analysis and detection of aqueous pharmaceuticals. However, before publication, there are also some suggestions for the authors to make moderate revisions.

Materials and Methods

1) Page 3 Line 109.

Perhaps you can explain here why can the signal represent the average effects from the whole samples?

 Ans: The beam spot of 3-4 mm of the THz beam on the sample and the eyedrop full cover on the Si sensor. It is a homogeneous signal at the irradiated site, rather than say an average signal across the sample. Revised.

2) Page 3 Line 116.

I wonder if methanol would affect the experimental results here, and could you explain why methanol is added? What are the benefits of doing so?

Ans:  The addition of methanol is to increase the dispersion properties of the mixed suspension of GO and CNT. After the coating is completed, the water and methanol are completely evaporated. There is no residue to affect the experimental results.

3) Page 4 Figure 1(b).

The two curves in the graph have a break around 0.1 THz, and for the black curve, there is a missing data portion of absorbance after approximately 2.6 THz. It is recommended to redraw the graph.

 Ans: Thanks, we redraw the graph.

4) Page 5 Figure 2.

Here I suggest that Figure 2 should add a legend to clearly differentiate between different curves.

 Ans: Yes, we export new figure in here.

5) Page 6 Figure 3.

The four images here do not correspond to the subsequent text descriptions. I suggest labeling each of these four images separately to facilitate clearer descriptions in the following text.

 Ans: Yes, we label and remark the scale of these SEM images.

6) Page 8 Line 223.

Could you provide a more detailed description of how the detector thickness affects the signal and explain the mechanisms behind it?

 Ans:  High resistance silicon substrate 400 µm. There is a refractive index solution between the silicon substrate and the silicon prism to reduce the loss of terahertz waves. Terahertz waves can penetrate the high-resistance silicon substrate and reach the upper interface between the eye drop and the sensor.

About the figure issues:

I have noticed that the figures in your paper are not very clear. I suggest exporting them as vector graphics for improving the quality of them.

Ans, Thanks remind these, we export all the figures with higher quality and more clearly.

Reviewer 3 Report

In the manuscript, the authors reported THz spectroscopy measurements on a commercial eye-drop liquid from Rohto pharmaceutical company. The main aim of the paper would be to test a method able to monitor aging and contamination effects on eye-drop samples. In this aim, the authors used hybrid graphene oxide and carbon nanotube nanostructures for improving the signal collection and enhancing the THz signal. The implemented spectroscopy method is valid. However, the manuscript has many failures from the point of view of methodology and presentation.
The title is not really explicative because the words “ the eye drop detection” are not appropriate. The THz should be used to monitor possible contamination effects. Considering the title it seems that the method can detect just the presence of eye drops.
The authors should discuss what information THz spectroscopy can give on deterioration and contamination of the sample, and the sensitivity and specificity characteristics of the method for practical use. These points should be discussed.
The comment reported on lines 129-141 is too much generic. The refs. 15 and 16 are not appropriately cited because they refer to the isotopic characterization of the water and FT-IR spectroscopy.
The sentence “In this work, we demonstrate enhanced THz spectra response with high sensitivity to the active ingredient in a commercial eye drop” is not experimentally supported.
The Raman spectroscopy and SERS have been completely neglected in the discussion, but these spectroscopy methods can provide efficacious ways to characterize liquid samples as eye drops (see, for instance, Y. Ding et al., Sensors 17 (2017) 2962; M. Elshout et al, J Ocul Pharmacol Ther. 27(5) (2011) 445.)
In the text, from lines 45 to 51, a list of titles is reported but their meanings are unclear.
The use of graphene-based nanomaterials to improve the THz spectroscopy should be better motivated and correlated to suitable references if it is the case.
No clear statements on the relevance and novelty of the proposed spectroscopy method have been done and discussed. Moreover, the citations are not exhaustive and, in some cases, inadequate.

For these reasons, I recommend rejecting the submission of the manuscript.

Author Response

Review 3

In the manuscript, the authors reported THz spectroscopy measurements on a commercial eye-drop liquid from Rohto pharmaceutical company. The main aim of the paper would be to test a method able to monitor aging and contamination effects on eye-drop samples. In this aim, the authors used hybrid graphene oxide and carbon nanotube nanostructures for improving the signal collection and enhancing the THz signal. The implemented spectroscopy method is valid. However, the manuscript has many failures from the point of view of methodology and presentation.

  • The title is not really explicative because the words “ the eye drop detection” are not appropriate. The THz should be used to monitor possible contamination effects. Considering the title it seems that the method can detect just the presence of eye drops.

Ans: Detailed eyedrop analysis has been added in.

THz spectral response and signal enhancement study of nano-sensor that can be used for eye drop detection

  • The authors should discuss what information THz spectroscopy can give on deterioration and contamination of the sample, and the sensitivity and specificity characteristics of the method for practical use. These points should be discussed.

  • Ans: We added more measurement information and equation to explain our method, and give more detail discussion in text.

  • The comment reported on lines 129-141 is too much generic. The refs. 15 and 16 are not appropriately cited because they refer to the isotopic characterization of the water and FT-IR spectroscopy.

  • Ans: the content in lines 129-141 has been modified and ref 15 and 16 have been replaced.

  • The sentence “In this work, we demonstrate enhanced THz spectra response with high sensitivity to the active ingredient in a commercial eye drop” is not experimentally supported.
    The Raman spectroscopy and SERS have been completely neglected in the discussion, but these spectroscopy methods can provide efficacious ways to characterize liquid samples as eye drops (see, for instance, Y. Ding et al., Sensors 17 (2017) 2962; M. Elshout et al, J Ocul Pharmacol Ther. 27(5) (2011) 445.)

  • Ans: Raman and SERS methods have been added in to the introduction part and overall the introduction part has been modified and enhanced.
  •  

  • In the text, from lines 45 to 51, a list of titles is reported but their meanings are unclear.
    The use of graphene-based nanomaterials to improve the THz spectroscopy should be better motivated and correlated to suitable references if it is the case.

  • Ans: content from line 45 to 51 has been modified and overall introduction has been improved. graphene-based nanomaterials to improve the THz spectroscopy has been added in and more references have been inserted

  • No clear statements on the relevance and novelty of the proposed spectroscopy method have been done and discussed. Moreover, the citations are not exhaustive and, in some cases, inadequate.

  • Ans: the novelty is with the enhanced thin film of graphene oxide/CNT, different types of eyedrop can be differentiated. There are detailed absorption peak information can be reviewed for each different type of eyedrop. The Graphene oxide is low cost and easy to fabricate.

Round 2

Reviewer 1 Report

All of my comments are all answered but not in detail. I suggest the authors to make more explanation about their replies to all of these three reviewers.

Reviewer 3 Report

The authors provided a revised version of the manuscript, but the changes in the text are not convincing answers to the indicated criticisms.
The first version of the manuscript has many failures that the revision did not overcome from the point of view of methodology and presentation.
I still believe the manuscript does not fit the requirements for publication in Sensors.